# Who Wrote it and Why?
# Prompting Large-Language Models for Authorship Verification

**Chia-Yu Hung,  Zhiqiang Hu,  Yujia Hu,  Roy Ka-Wei Lee**

Singapore University of Technology and Design

{chiayu_hung, zhiqiang_hu}@mymail.sutd.edu.sg

{yujia_hu, roy_lee}@sutd.edu.sg

## Abstract

Authorship verification (AV) is a fundamental task in natural language processing (NLP) and computational linguistics, with applications in forensic analysis, plagiarism detection, and identification of deceptive content. Existing AV techniques, including traditional stylometric and deep learning approaches, face limitations in terms of data requirements and lack of explainability. To address these limitations, this paper proposes PromptAV, a novel technique that leverages Large-Language Models (LLMs) for AV by providing step-by-step stylometric explanation prompts. PromptAV outperforms state-of-the-art baselines, operates effectively with limited training data, and enhances interpretability through intuitive explanations, showcasing its potential as an effective and interpretable solution for the AV task.

## 1 Introduction

**Motivation.** Authorship verification (AV) is a fundamental task in the field of natural language processing (NLP). It aims to determine the authorship of a given text by analyzing the stylistic characteristics and patterns exhibited in the writing (Stamatatos, 2016). The importance of AV lies in its wide range of applications, including forensic analysis (Iqbal et al., 2010), plagiarism detection (Stein et al., 2011), and the detection of deceptive or fraudulent content (Claxton, 2005). The rise of Large-Language Models (LLMs) (Chowdhery et al., 2022; Brown et al., 2020), has facilitated the generation of human-like text where one would now need to factor in the possibility of machine-generated text (Uchendu et al., 2023). This development give rise to a new layer of complexity in distinguishing between machine-generated and human-written text, subsequently amplifying the significance of the AV problem.

Over the years, various techniques have been proposed to address the AV task. Traditional approaches relied on stylometric features, such as n-gram frequencies, vocabulary richness, and syntactic structures, to capture the unique writing style of individual authors (Ding et al., 2017). Machine learning techniques, such as support vector machines (SVMs) and random forests, have been employed to model the relationship between these stylometric features and authorship (Brocardo et al., 2013). More recently, deep learning approaches have shown promising AV results by learning intricate patterns directly from raw text data (Bagnall, 2015; Ruder et al., 2016; Fabien et al., 2020).

Despite these advancements, existing AV techniques have certain limitations. Firstly, most existing AV methods require a large amount of labeled training data, which can be costly to acquire, especially for scenarios with limited available data. Secondly, there is a lack of explainability in the predictions made by these techniques. In many practical applications, it is crucial to understand why a particular decision was reached, which is particularly relevant in legal contexts or situations where interpretability is paramount.

**Research Objectives.** To address these limitations, We propose PromptAV, a novel technique that utilizes LLMs for authorship attribution by employing step-by-step stylometric explanation prompts. These prompts guide the LLMs to analyze and evaluate the textual features that contribute to authorship, enabling the model to provide explanations for its predictions. PromptAV demonstrates effectiveness in both zero-shot and few-shot settings, offering improved performance compared to existing baselines, especially in scenarios with limited training data. Additionally, detailed case studies showcase PromptAV's ability to provide intuitive explanations for its predictions, shedding light on the factors contributing to authorship attribution and enhancing the interpretability of the AV process.

**Contribution.** This paper introduces PromptAV, a prompt-based learning technique that har-

**Chain -of -Thought prompting**

(a)

> Task: *On a scale of 0 to 1, with 0 indicating low confidence and 1 indicating high confidence, please provide a general assessment of the likelihood that Text 1 and Text 2 were written by the same author. Your answer should reflect a moderate level of strictness in scoring.* Let's think step by step.
>
> Text 1: [T1]
> Text 2: [T2]

**PS+ prompting**

(b)

> Task: *On a scale of 0 to 1, with 0 indicating low confidence and 1 indicating high confidence, please provide a general assessment of the likelihood that Text 1 and Text 2 were written by the same author. Your answer should reflect a moderate level of strictness in scoring. First step: Understand the problem, extracting relevant variables and devise a plan to solve the problem. Then, carry out the plan and solve the problem step by step. Finally, show the confidence score.*
>
> Text 1: [T1]
> Text 2: [T2]

**PromptAV**

(c)

> Task: *On a scale of 0 to 1, with 0 indicating low confidence and 1 indicating high confidence, please provide a general assessment of the likelihood that Text 1 and Text 2 were written by the same author. Your answer should reflect a moderate level of strictness in scoring.* Here are some relevant variables to this problem.
> 1. punctuation style(e.g. hyphen, brackets, colon, comma, parenthesis, quotation mark)
> 2. special characters style, capitalization style(e.g. Continuous capitalization, capitalizing certain words)
> 3. acronyms and abbreviations(e.g. Usage of acronyms such as OMG, Abbreviations without punctuation marks such as Mr Rochester vs. Mr. Rochester,Unusual abbreviations such as def vs. definitely)
> 4. writing style
> 5. expressions and Idioms
> 6. tone and mood
> 7. sentence structure
> 8. any other relevant aspect
> *First step: Understand the problem, extracting relevant variables and devise a plan to solve the problem. Then, carry out the plan and solve the problem step by step. Finally, show the confidence score.*
>
> Text 1: [T1]
> Text 2: [T2]

Figure 1: Prompts used by (a) CoT, (b) PS+ prompting, (c) PromptAV for a AV task. The trigger intructions of the various techniques are highlighted in the prompt.

nesses the linguistics capabilities of LLMs for AV. By providing step-by-step stylometric explanation prompts, PromptAV achieves improved performance, offers explainability in its predictions, and operates effectively in both zero-shot and few-shot settings. The experimental results demonstrate the potential of PromptAV as an effective and interpretable solution for the AV task.

## 2 Related Work

AV is a well-established research topic. Classical approaches to AV have predominantly centered on stylometric features encompassing n-gram frequencies, lexical diversity, and syntactic structures to discern the unique writing styles intrinsic to individual authors (Stamatatos, 2016; Lagutina et al., 2019). Classical machine learning algorithms were also used to model the associations between these stylometric attributes and authorship (Brocardo et al., 2013; Hu et al., 2023). Recently, deep learning models such as RNNs and CNNs being employed to extract more complex patterns from textual data (Bagnall, 2015; Ruder et al., 2016; Benzebouchi et al., 2018). These deep learning architectures have exhibited promising advancements in AV tasks. Furthermore, the advent of the

Transformer architecture (Vaswani et al., 2017) has prompted the development of Transformer-based AV models (Ordoñez et al., 2020; Najafi and Tavan, 2022; Tyo et al., 2021).

## 3 Methodology

In an recent work, Wei et al. (2023) have proposed *Chain of thought* (CoT) prompting, which employs a series of intermediate reasoning steps to significantly improve the ability of LLMs in performing complex reasoning. Wang et al. (2023) further extended the concept to propose *PS+ prompting*, a zero-shot prompting methodology that instructs LLMs to formulate a strategic plan prior to problem-solving.

Inspired by these works, we introduce PromptAV, a prompting strategy that incorporates key linguistic features—identified in (Boenninghoff et al., 2019) as the crucial variables for LLMs to evaluate. These linguistic features serve as rich, often subtle, markers of an author's distinct writing style and are hence of paramount importance for the AV task. For instance, the consistent use or avoidance of certain punctuation marks, or the preference for particular special characters, can be idiosyncratic to an author. Similarly, elements

like tone and expression, though harder to quantify, can offer significant insights into an author's unique 'voice'. The integration of these linguistic attributes within the PromptAV framework does not simply add layers of complexity to the model, it fundamentally enhances the model's ability to capture the author-specific nuances within textual data. This capability, in turn, improves the model's accuracy and reliability when attributing authorship to unidentified texts, making PromptAV a powerful tool in the field of AV. Figure 1 provides an overview that compares the prompts used in CoT, PS+ prompting, and PromptAV.

We also note that conventional prompting techniques that instruct the LLM to respond strictly with binary "*yes*" or "*no*" answers frequently result in the LLM responding '*no*' for the majority of instances within an AV task. To mitigate this problem, PromptAV instructs the LLM to generate a confidence score ranging from 0 to 1, rather than a binary response. To calibrate these generated confidence scores, we augment the prompt with the additional directive "*Your answer should reflect a moderate level of strictness in scoring*".

## 3.1 Few Shot Prompting

In the conventional setup of few-shot CoT prompting, each example necessitates the manual crafting of reasoning steps, an approach that is impractical for the AV task. This is primarily due to the challenges in ensuring the consistent quality of hand-crafted reasoning across diverse prompting templates. Hence, we resort to leveraging the capacity of LLMs as zero-shot reasoners to generate the required reasoning steps (Kojima et al., 2023).

To construct the few-shot PromptAV and CoT examples, we introduce an additional directive to the original prompt: "*It is given that after following the instruction, the confidence score obtained is [X]. Show the step-by-step execution of the instruction so that this score is achieved.*" Here, *[X]* is either 0.9 or 0.1, contingent upon whether the pair of texts were written by the same author. Subsequently, we feed this modified prompt to GPT-4 [1] to generate intermediate reasoning steps.

Zhao et al. (Zhao et al., 2021) has demonstrated that both the selection of training examples and their sequencing can significantly influence accuracy. Therefore, to ensure fairness, we utilized the same examples for all our 2-shot and 4-shot experiments. These examples were randomly sampled from the training set and presented to the LLM in the same order.

| Setting | Model | Accuracy |
|---------|-------|----------|
| 0-Shot | PromptAV | **0.587** |
| | PS+ prompting | 0.536 |
| | CoT | 0.524 |
| 2-Shot | PromptAV | **0.667** |
| | CoT | 0.595 |
| 4-shot | PromptAV | **0.635** |
| | CoT | 0.510 |

Table 1: Performance of PromptAV and the baselines on *k*-shots settings. Highest accuracy are **bolded**.

## 4 Experiments

### 4.1 Experimental Settings

**Dataset.** We evaluate PromptAV and the baselines using IMDb62 (Seroussi et al., 2014), which is commonly used dataset for AV task. Each record in dataset contains a pair of texts and the ground truth labels, i.e., a "*positive*" pair of texts belong to the same author, and "*negative*" for otherwise. As we are interested in performing the AV task in zero-shot and few-shot settings, we focus on sampling a manageable test set of 1,000 text pairs from the original IMDb62 dataset. The resulting test set comprises 503 positive and 497 negative text pairs. **Implementation.** All of our experiments were conducted on the `gpt-3.5-turbo` model, with the temperature parameter set to 0. We benchmarked PromptAV against *PS+ prompting* in zero-shot settings and against *CoT* prompting in zero, two, and four-shot settings. Given that all three methods generate a confidence score in the range of 0 to 1, we reported the accuracy corresponding to the optimal threshold that yields the highest accuracy. We call this the Optimal Threshold Methodology and we will dicuss more about it in Appendix A.3.

### 4.2 Experimental Results

Table 1 shows the performance of PromptAV and the baselines on *k*-shot settings. We observe that PromptAV consistently achieved higher accuracy across varying shot settings. This demonstrate PromptAV's efficacy in performing AV task with little or no training data. Furthermore, as the number of shots increases from 0 to 4, PromptAV maintains its edge over the CoT. This suggests that PromptAV effectively leverages the additional information provided in the few-shot scenarios, optimizing its decision-making process.

---

[1] https://openai.com/research/gpt-4

| PromptType | Generated Explanation |
|---|---|
| | **Text 1**: *" We're all sinners , or about to be . " I picked up this flick as part of a double film DVD offer along with " The Hanged Man " for just a buck , and truth be told , both movies were a whole lot better than either had a right to be for the price . The title on this one was " The Gun and The Cross " , close enough to the original to get the message across . The title character wields both , as Marjoe Gortner is revealed to be gunfighter Ernie Parsons , masquerading as a preacher after he finds a real one who was bushwhacked . Donning the clothes of the dead minister , he arrives at the town of Castle Walk just in time to help the local folk deal with nasty town boss Ross ( David Huddleston ) ...* [Truncated due to length restriction] |
| | **Text 2:** *" This is a mad dream Antinea , but I shall put an end to the madness . " The title " Hercules and the Captive Women " is a bit of a misnomer , there was only one woman captive , and along with her mother , Queen Antinea of Atlantis , were the only two women in the entire film . The International English title mentions the ' Conquest of Atlantis ' , and that probably makes more sense , but in the end , it's the destruction of Atlantis that takes center stage . There's an entire mythology offered by Zantas , the last high priest of Uranus , that explains what's going on with Antinea and her cohort of Grand Vizier clones . ...* [Truncated due to length restriction] |

| PromptType | Generated Explanation |
|---|---|
| 0-shot PromptAV | Based on the provided texts, here is a general assessment of the likelihood that Text 1 and Text 2 were written by the same author:
1. Punctuation style: Both texts use similar punctuation styles, such as commas, periods, and quotation marks. However, Text 1 uses more parentheses and dashes compared to Text 2. Confidence: 0.7
2. Special characters style, capitalization style: Both texts have similar capitalization styles, with proper nouns and the beginning of sentences capitalized. However, Text 1 has more continuous capitalization (e.g., "The Gun and The Cross") compared to Text 2. Confidence: 0.6
3. Acronyms and abbreviations: Both texts use acronyms and abbreviations, but there are no significant differences in their usage. Confidence: 0.8
4. Writing style: Both texts have a similar writing style, with a mix of descriptive and narrative elements. However, Text 1 focuses more on reviewing a movie, while Text 2 provides a summary of a film's plot. Confidence: 0.7
5. Vocabulary usage: Both texts use a variety of vocabulary, but there are no significant differences in their usage. Confidence: 0.8
Overall, based on the assessment of the relevant variables, there is a moderate likelihood (confidence score: 0.7) that Text 1 and Text 2 were written by the same author. |
| 0-shot CoT | Based on a moderate level of strictness in scoring, the likelihood that Text 1 and Text 2 were written by the same author is low. The two texts have different subject matters and writing styles, with Text 1 discussing a western film and Text 2 discussing a Hercules film. Additionally, the tone and language used in each text are distinct. Therefore, it is unlikely that the same author wrote both texts.
Based on these factors, I would assess the likelihood that Text 1 and Text 2 were written by the same author as low, with a confidence level of around 0.3 on the scale of 0 to 1. |
| 0-shot PS+ prompting | Based on a moderate level of strictness in scoring, I would assess the likelihood that Text 1 and Text 2 were written by the same author as low. The two texts have different themes, writing styles, and subject matters. Text 1 discusses a western film, while Text 2 discusses a Hercules film. The language and tone used in each text are also distinct. Therefore, I would assign a confidence score of 0.2, indicating a low likelihood of the same authorship. |

Table 2: A text pair example with the generated responses from PromptAV, CoT and PS+ Prompting

| Model | Accuracy |
|---|---|
| PromptAV | **0.587** |
| PromptAV_Obfuscated | 0.580 |
| PS+ prompting | 0.536 |
| CoT | 0.524 |

Table 3: Performance of PromptAV against a series of authorship obfuscation methods. Highest accuracy are **bolded**

The superior performance of PromptAV over PS+ prompting and CoT in the zero-shot setting underscores its capacity to make effective use of key linguistic features for AV even in the absence of training examples. This is indicative of the fact that PromptAV is adept at understanding intrinsic linguistic characteristics, which are critical in differentiating writing styles of various authors.

### 4.3 Explainability Case Studies

While achieving explainability in the AV task is notoriously difficult, PromptAV rises to the challenge by providing detailed, interpretable solutions. This is accomplished through a meticulous analysis of the linguistic features. A sample of the generated responses from PromptAV, CoT, and PS+ prompting is illustrated in Table 2. In contrast to CoT prompting or PS+ prompting, PromptAV offers a comprehensive comparison of linguistic features, such as punctuation style and vocabulary usage. For example, PromptAV detects that both

texts have similar capitalization style and writing style, with a mix of descriptive and narrative elements. On the other hand, CoT and PS+ prompting systems deliver a more superficial level of analysis. While they can recognize discrepancies in vocabulary selection across texts, they lack the capacity to provide an exhaustive explanation or a deeper analysis of these differences.

## 4.4 Authorship obfuscation

In our pursuit to understand PromptAV's potential robustness, we took a focused approach by testing it against an obfuscation dataset. We selected the suggested Mutant-X algorithm (Mahmood et al., 2020), a recognized method in the authorship obfuscation domain, and applied it to the IMDb62 dataset that was referenced in our paper. We apply PromptAV in zero-shot setting on the obfuscated dataset and benchmark against its performance on the original dataset. The experimental results are presented in Table 3. The results reveal a negligible decline in performance when confronted with obfuscated text. Interestingly, the accuracy achieved, even with obfuscation, surpasses the results of zero-shot CoT and PS+ prompting on non-obfuscated text. This indicates a promising level of resilience in PromptAV against authorship obfuscation methods. While these preliminary results are encouraging, we acknowledge the need for more comprehensive testing against a broader range of obfuscation methods to assert PromptAV's robustness conclusively.

## 4.5 Stylometric Feature Impact

The field of stylometry is vast, with a plethora of features available for authorship analysis. The central aim of PromptAV was to showcase the flexibility and adaptability of the framework when working with different sets of stylometric features. We conducted a series of experiments using varying sets of stylometric features with PromptAV in zero-shot setting. We report the result in Table 4. The result clearly showcases that the performance varies based on the chosen feature set. The peak accuracy achieved with a 9-feature set emphasizes that PromptAV's success is contingent on the quality and appropriateness of the selected features. We genuinely believe that the field holds significant undiscovered potential, with combinations of features that could further elevate the performance of PromptAV. Our results only scratch the surface, and our intention is to spur the community into exploring these myriad possibilities.

## 5 Conclusion

In this paper, we proposed PromptAV, a prompt-based learning technique that harnesses the linguistics capabilities of LLMs for AV. Our experimental results demonstrated PromptAV's superior performance over the state-of-the-art baselines in zero-shot and few-shot settings. Case studies were also conducted to illustrate PromptAV's effectiveness and explainability for the AV task. For future works, we will conduct experiments on more AV datasets and explore other linguistic features.

## 6 Limitations

This study, while substantive, acknowledges certain limitations. A significant constraint is the absence of a ground truth for the generated explanation, which renders the evaluation of the efficacy of PromptAV's generated explanation somewhat challenging. This suggests a potential avenue for tapping into the expertise of forensic linguistic experts to assess and curate some of the reasoning generated by PromptAV. Additionally, we have observed instances where PromptAV generates illusory explanations by identifying the usage of specific vocabulary within the text, even when such vocabulary may not have been actually used.

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

## A    Appendix

### A.1    Computation and Time comsumption of PromptAV

In our experiments, we applied PromptAV using GPT-3.5 via the OpenAI API. Specifically, evaluating 1,000 examples took approximately 8 hours. The expense associated with using PromptAV via the OpenAI API is indeed calculated at a rate of $0.004 per 1,000 tokens. For a dataset of 1,000 samples, the estimated cost came to around $10. This cost, we believe, is justifiable given the unique interpretability insights PromptAV provides, especially in critical applications like forensic analysis where the implications of the results are profound. It is possible to batch-process texts through sending multiple requests using multiple API keys. With adequate budget, we could scale-up the AV operations.

### A.2    Stylometric Feature Impact

| Number of features | Accuracy |
|---|---|
| 8 | 0.587 |
| 9 | **0.602** |
| 10 | 0.585 |
| 11 | 0.564 |
| 12 | 0.572 |

Table 4: Performance of PromptAV in 0 shot setting with varying sets of stylometric features.

### A.3    Optimal Threshold Methodology

The method of reporting using an optimal threshold was our initial approach to compare the results of different prompting methods. The intention was to establish a baseline comparison across varying methods in a controlled environment. In real-world applications, the threshold would likely require adjustment based on specific use-cases and available data.

Practical Implications of Threshold: It is indeed a challenge to select an optimal threshold that would be universally valid. In our experiment applying PromptAV on the IMDb62 dataset, we found that the optimal threshold lies between 0.2 - 0.3. This range suggests that the model tends to generate a low confidence score, which provides significant insights into its functioning and decision-making. The low confidence score may be indicative of the

| Confidence Score | PromptAV | PS+ | CoT |
|---|---|---|---|
| 0.1 | 0.506 | 0.501 | **0.524** |
| 0.2 | **0.587** | **0.536** | 0.522 |
| 0.3 | 0.579 | 0.529 | 0.505 |
| 0.4 | 0.578 | 0.529 | 0.497 |
| 0.5 | 0.569 | 0.527 | 0.497 |
| 0.6 | 0.55 | 0.484 | 0.498 |
| 0.7 | 0.53 | 0.493 | 0.498 |
| 0.8 | 0.519 | 0.497 | 0.497 |
| 0.9 | 0.5 | 0.497 | 0.497 |

Table 5: Performance of PromptAV and the baselines in 0 shot setting with varying confidence score threshold. Highest accuracy are bolded

model's cautious approach to attributing authorship, ensuring a reduced number of false positives in practical scenarios. This threshold may varied, depending on the complexity of the dataset, and the amount of writing sample observations the model made for each author.

Selection of the Confidence Score: The confidence score selection was an empirical decision based on our preliminary experiments, intending to optimize the balance between false positives and false negatives. We have conducted an experiments to show the tradeoff on performance when selecting different threshold for the confident score. Note that experiments are conducted using PromptAV with zero-shot demonstration setting.

### A.4    Zero-shot vs supervised methods

Our work with PromptAV centers around its capability to handle the AV task in a zero-shot setting. Acquiring labeled data, especially for tasks like AV, can be quite challenging due to privacy concerns, the need for expert annotators, and other complexities associated with the domain. We recognize the importance and relevance of benchmarking against state-of-the-art AV methods. Their dependence on copious amounts of labeled data inherently places them in a different operating paradigm. For instance, BERT is able to achieve a high accuracy of 0.702 if it is trained on IMDb62 full training dataset with 74,400 samples. However, BERT will require large amount of training data to achieve the superior performance.