# OpenReview forum: "Who Wrote it and Why? Prompting Large-Language Models for Authorship Verification"
_EMNLP/2023/Conference — EMNLP 2023 Findings_

### Official Review · Reviewer_ug9M · 2023-08-04

**Soundness:** 3

**Excitement:**

4: Strong: This paper deepens the understanding of some phenomenon or lowers the barriers to an existing research direction.

**Paper Topic And Main Contributions:**

The paper introduces an approach for prompting large language models (GPT 3.5 in this case) to evaluate stylistic similarity between two text samples for authorship verification (AV). The prompt used in the paper guides the language models towards using a pre-determined set of known stylometric features for evaluation culminating in the generation of a confidence score. This confidence score is thresholded to decide if the samples belong to the same author. The output generated includes human-comprehensible reasoning for interpretability. The prompting introduced in this paper, called PromptAV, is compared to existing prompt-based methods using zero-shot and few-shot evaluation protocols on data from IMDB62 (a common AV corpus). Effective accuracy ranges from 59% for zero-shot to a maximum of 67% for few-shot learning.

**Questions For The Authors:**

1. Was the threshold for the confidence score fixed for all three methods?

**Reasons To Accept:**

Prompting language model for assessing stylistic similarity between text samples is a fresh perspective on AV. Leveraging the vast capabilities of large language models without corpus-specific training (as demonstrated through zero-shot and few-shot experiments) demonstrates significant potential. Further, the human comprehensible explanation in natural language may be very useful in legal and forensic contexts especially to users not well-versed in AI methods.

**Reasons To Reject:**

There are two weaknesses in the paper. First, the reliance on the pre-determined set of stylometric features. There are a lot of stylometric features in use in the community and common consensus on utility only holds for a few of them. So, this limitation may be detrimental. Second, the method was only compared with existing prompting methods (chain-of-through and PS+prompting). Comparing this approach to the state-of-the-art methods in AV is necessary to truly understand the utility and potential of the proposed approach.

**Reproducibility:**

3: Could reproduce the results with some difficulty. The settings of parameters are underspecified or subjectively determined; the training/evaluation data are not widely available.

**Reviewer Confidence:**

4: Quite sure. I tried to check the important points carefully. It's unlikely, though conceivable, that I missed something that should affect my ratings.

**Typos Grammar Style And Presentation Improvements:**

Page 2 line 106 steps to significantly improves the ability
Page 3 line 136 PS+promptingm
Page 3 line 203 textbfPromptAV

---

> ### Author Rebuttal · Authors · 2023-08-29
>
> Thank you for your interesting questions. Your thought-provoking questions have significantly contributed to enhancing our paper's quality. We hope that we have comprehensively addressed the concerns and questions you presented. Based on the valuable discussion, we will make the necessary updates to our paper.
>
> ---
>
> ***W1**: First, the reliance on the pre-determined set of stylometric features. There are a lot of stylometric features in use in the community and common consensus on utility only holds for a few of them. So, this limitation may be detrimental.*
>
> **[Response to W1]**
>
> We fully acknowledge that the field of stylometry is vast, with a plethora of features available for authorship analysis. It's indeed true that a consensus on the utility of many of these features remains elusive. The central aim of PromptAV was not to assert that a particular set of features are universally superior, but rather to showcase the flexibility and adaptability of the framework when working with different sets of stylometric features.
>
> PromptAV's design allows it to be feature-agnostic, meaning it has the potential to accommodate any feature set that researchers or practitioners might find relevant. This flexibility ensures that as the domain of stylometry evolves, and as new insights about feature utility emerge, PromptAV can continue to adapt and remain relevant.
>
> To underscore this adaptability, we conducted a series of experiments using varying sets of stylometric features with PromptAV in zero-shot setting. The data clearly showcases that the performance varies based on the chosen feature set. The peak accuracy achieved with a 9-feature set emphasizes that PromptAV's success is contingent on the quality and appropriateness of the selected features.
>
> | Number of features|Accuracy|
> |-|-|
> |8| 0.587 |
> |9 | 0.602 |
> |10| 0.585 |
> |11| 0.564 |
> |12| 0.572 |
>
> We genuinely believe that the field holds significant undiscovered potential, with combinations of features that could further elevate the performance of PromptAV. Our results only scratch the surface, and our intention is to spur the community into exploring these myriad possibilities. We will actively encourage the community to explore various feature sets, and we hope that collectively, we can push the boundaries of what is achievable in authorship verification using PromptAV.
>
> ---
>
> ***W2**: Second, the method was only compared with existing prompting methods (chain-of-through and PS+prompting). Comparing this approach to the state-of-the-art methods in AV is necessary to truly understand the utility and potential of the proposed approach.*
>
> **[Response to W2]**
>
> Your point about the comparison with only existing prompting methods is valid. Our primary aim was to juxtapose PromptAV with methods that operate under similar conditions and constraints. The methods we chose for comparison, namely the chain-of-thought and PS+ prompting, align closely with the premise and operating principles of PromptAV.
>
> We certainly recognize the importance and relevance of benchmarking against state-of-the-art AV methods. However, the vast majority of these methods are supervised, leveraging large volumes of labeled training data to achieve their results. Their dependence on copious amounts of labeled data inherently places them in a different operating paradigm. For instance, BERT is able to achieve a high accuracy of 0.702 if it is trained on IMDb62 full training dataset with 74,400 samples. However, BERT will require large amount of training data to achieve the superior performance.
>
> In our study, we deliberately focused on a zero-shot AV setting, which, as you might agree, presents a uniquely challenging landscape. PromptAV, as well as the other prompting baselines we evaluated, operates without the benefit of labeled training data, relying instead on the generalization capabilities of language models. The essence of our research was to probe the extent to which such methods could navigate the complexities of the AV task in the absence of explicit training on it. Furthermore, it is essential to highlight that PromptAV focuses not just on accuracy but also on providing detailed step-by-step stylometric explanations.
>
> However, we acknowledge the value of your feedback and recognize that benchmarking PromptAV's performance against state-of-the-art methods, even with the stated caveats, could offer additional layers of insights into its efficacy and potential areas of enhancement. We will include a discussion on supervised and zero-shot AV task in our updated version, and report the performance of some of the state-of-the-art supervised AV methods.
>
>  ---
>
> ***Q1**: Was the threshold for the confidence score fixed for all three methods?*
>
> **[Response to Q1]**
>
> To clarify, the threshold for the confidence score was not held constant across all methods. Instead, based on empirical observations, we determined specific optimal thresholds for each method to achieve the best performance. For instance, while we set the threshold for PromptAV at 0.2, the thresholds for PS+ prompting and CoT were fixed at 0.2 and 0.1 respectively. Note that this is for the zero-shot settings.
>
> We have also responded to a similar question by Reviewer 4xL7. Specifically, we utilized the optimal thresholding approach primarily to facilitate a straightforward comparative assessment across the different prompting techniques. By doing so, we could establish a foundation for understanding each method's inherent strengths and weaknesses in a controlled setting. However, we acknowledge that in real-world applications, determining the ideal threshold could necessitate further tailoring to meet the intricacies of specific datasets and LLMs.
>
> The derivation of the confidence score was empirically driven, grounded in preliminary experimentation. Our aim was to strike a judicious balance between false positives and false negatives. We have conducted an experiments to show the tradeoff on performance (in terms of accuracy) when selecting different threshold for the confident score for the various models. Note that experiments are conducted in zero-shot setting. We will include this experiment results in the appendix of our camera-ready version.
>
> | #Confidence Score Threshold | PrompAV | PS+ | CoT |
> |-|-|-|-|
> | 0.1|0.506|0.501|0.524|
> | 0.2|0.587|0.536|0.522|
> | 0.3|0.579|0.529|0.505|
> | 0.4|0.578|0.529|0.497|
> | 0.5|0.569|0.527|0.497|
> | 0.6|0.55|0.484|0.498|
> | 0.7|0.53|0.493|0.498|
> | 0.8|0.519|0.497|0.497|
> | 0.9|0.5|0.497|0.497|
>
> ---
>
> ***Others**: Grammar Issues*
>
> Thank you for pointing out the grammar mistakes in the paper. We have corrected them, and we will perform a more thorough proofreading in our camera-ready version.

---

### Official Review · Reviewer_Waug · 2023-08-06

**Soundness:** 3

**Excitement:**

3: Ambivalent: It has merits (e.g., it reports state-of-the-art results, the idea is nice), but there are key weaknesses (e.g., it describes incremental work), and it can significantly benefit from another round of revision. However, I won't object to accepting it if my co-reviewers champion it.

**Paper Topic And Main Contributions:**

The paper studies authorship verification (AV) and focuses on two problems: 1. data requirements and 2. lack of explainability. The paper proposes PromptAV to leverage LLM (GPT-3.5) and provide step-by-step stylometric explanation prompts. Specifically, PromptAV instructs LLM with linguistic attributes in order to capture author-specific nuances. The paper shows that PromptAV outperforms baselines on 0-short, 2-short, and 4-short settings.

**Questions For The Authors:**

Do you know if GPT-3.5 sees IMDb62 during training?

**Reasons To Accept:**

1. Introducing linguistic features as prompts for AV seems to be novel.
2. PromptAV yields better performance than baselines.

**Reasons To Reject:**

1. It is not clear how PromptAV could address the limitation of data requirements since it is mentioned in the abstract and introduction. The paper did not provide a solution for saving data usage. If the justification is to use GPT-3.5 instead of training a model, one could argue that the paper actually requires more data as it depends on the performance of LLM.


**Reproducibility:**

4: Could mostly reproduce the results, but there may be some variation because of sample variance or minor variations in their interpretation of the protocol or method.

**Reviewer Confidence:**

4: Quite sure. I tried to check the important points carefully. It's unlikely, though conceivable, that I missed something that should affect my ratings.

---

> ### Author Rebuttal · Authors · 2023-08-29
>
> Thank you for your interesting questions. Your thought-provoking questions have significantly contributed to enhancing our paper's quality. We hope that we have comprehensively addressed the concerns and questions you presented. Based on the valuable discussion, we will make the necessary updates to our paper.
>
> ---
>
> ***W1**: It is not clear how PromptAV could address the limitation of data requirements since it is mentioned in the abstract and introduction. The paper did not provide a solution for saving data usage. If the justification is to use GPT-3.5 instead of training a model, one could argue that the paper actually requires more data as it depends on the performance of LLM.*
>
> **[Response to W1]**
>
> You've rightly pointed out the data-intensive nature of training LLMs. However, the essence of our work with PromptAV centers around its capability to handle the Authorship Verification (AV) task in a zero-shot setting.
>
> While it's true that models like GPT-3.5 have been pretrained on vast datasets, this is not an additional data requirement imposed by our approach. In fact, the primary advantage we highlight with PromptAV is its elimination of the need for a labeled AV-specific dataset for further training. As you may appreciate, acquiring labeled data, especially for tasks like AV, can be quite challenging due to privacy concerns, the need for expert annotators, and other complexities associated with the domain. Our solution alleviates this need by leveraging the power of pretrained LLMs and effectively guiding them through prompts to achieve competitive performance without further AV-specific training.
>
> In sum, while we acknowledge the extensive data needs of LLMs like GPT-3.5 in their initial training phases, PromptAV's contribution is to leverage these pretrained models for AV without necessitating additional labeled data, making it a practically feasible solution for many real-world scenarios.
>
> For future work, we will also explore evaluating PromptAV’s performance tradeoffs using LLMs of various sizes.
>
> ---
>
> ***Q1**: Do you know if GPT-3.5 sees IMDb62 during training?*
>
> **[Response to Q1]**
>
> We are not privy to the precise datasets OpenAI utilized in pretraining GPT-3.5. OpenAI hasn't expressly delineated the inclusion of IMDb62 in its pretraining corpus. Similarly, while we are aware that the authorship verification task isn't among the instruction tuning tasks (FLAN) offered by Google, we cannot categorically state whether OpenAI incorporated the authorship verification task in GPT-3.5's fine-tuning process.
>
> That said, there are strong empirical reasons supporting our belief that GPT-3.5 hasn’t encountered IMDb62. As you correctly noted, if GPT-3.5 was familiar with IMDb62, one would expect significantly better accuracy from methods such as zero-shot chain of thought prompting or PS+ prompting. Instead, these methods produce results roughly analogous to a random guess, implying GPT-3.5's unfamiliarity with the IMDb62 dataset.
>
> While this observation is indicative, we are striving to ensure our claims remain consistent with the data we have on hand. We believe it's vital to differentiate between what we empirically observe and what we can confirm categorically. This distinction will be made more explicit in the limitation section of our revised version.

---

### Official Review · Reviewer_4xL7 · 2023-08-09

**Soundness:** 4

**Excitement:**

3: Ambivalent: It has merits (e.g., it reports state-of-the-art results, the idea is nice), but there are key weaknesses (e.g., it describes incremental work), and it can significantly benefit from another round of revision. However, I won't object to accepting it if my co-reviewers champion it.

**Missing References:**

Asad Mahmood, Faizan Ahmad, Zubair Shafiq, Padmini Srinivasan, and Fareed Zaffar. 2019. A Girl Has No Name: Automated Authorship Obfuscation using Mutant-X. Proc. Priv. Enhancing Technol. 2019, 4 (2019), 54–71.

Adaku Uchendu, Thai Le, and Dongwon Lee. 2022. Attribution and Obfuscation of Neural Text Authorship: A Data Mining Perspective. arXiv preprint arXiv:2210.10488 (2022).

Maël Fabien, Esaú Villatoro-Tello, Petr Motlicek, and Shantipriya Parida. 2020. BertAA: BERT fine-tuning for Authorship Attribution. In Proceedings of the 17th International Conference on Natural Language Processing (ICON). 127–137.

Hope these papers can help you.

**Paper Topic And Main Contributions:**

This paper designs PromptAV, which adopts LLMs in solving a traditional NLP task, authorship verification, and uses chain-of-thought to prompt the verification process. With carefully designed prompts, LLMs can achieve about 60% accuracy on the IMDb62 dataset. PromptAV improves the traditional AV methods that need training data and models in supervised learning settings. It also confirms that LLMs can serve as tools for unsupervised learning in different domains.

**Questions For The Authors:**

According to the limitations above, there are the questions:

(1) Is PromptAV able to batch-process texts with reasonable time consumption, as AV or AA (authorship attribution) models do?

(2) The paper uses an optimal threshold when reporting accuracy. Is this valid in real application? Can the authors provide more insights in the selection of the confidence score?

(3) Is PromptAV robust against a series of authorship obfuscation methods? If so, why?

**Reasons To Accept:**

The topic of the paper, authorship verification, is interesting and meaningful in today's Internet and social media. Using new techniques in AV is a good attempt.

Zero-shot PromptAV proposed by the authors is suitable for AV in real scenarios, where people may not be able to get enough training data and train a model. PromptAV makes it possible for the masses to perform AV.

**Reasons To Reject:**

The topic of this article is intriguing, but I do agree that it falls short in terms of providing comprehensive experiments and further insights, making it insufficient for publication as an academic paper now. The authors should consider and address the following questions in further detail:

(1) The computation and time consumption of PromptAV is not analyzed.

(2) The practicality of PromptAV is not fully considered, especially in experiment part.

(3) The robustness of PromptAV is not analyzed, which is crucial for AV.

**Reproducibility:**

4: Could mostly reproduce the results, but there may be some variation because of sample variance or minor variations in their interpretation of the protocol or method.

**Reviewer Confidence:**

5: Positive that my evaluation is correct. I read the paper very carefully and I am very familiar with related work.

---

> ### Author Rebuttal · Authors · 2023-08-29
>
> Thank you for the insightful feedback and questions. Your thought-provoking questions have significantly contributed to enhancing our paper's quality. We hope that we have comprehensively addressed the concerns and questions you presented. Based on the valuable discussion, we will make the necessary updates to our paper.
>
> ---
>
> ***W1/Q1**: The computation and time consumption of PromptAV is not analyzed. Is PromptAV able to batch-process texts with reasonable time consumption, as AV or AA (authorship attribution) models do?*
>
> **[Response to W1/Q1]**
>
> We are glad you brought this up as this information is pivotal when discussing the practicality of our method. To provide clarity, our proposed technique, PromptAV, utilizes Large Language Models (LLMs) for authorship verification by employing step-by-step stylometric explanation prompts. The computation and time consumption is dependent on the LLMs used.
>
> In our experiments, we applied PromptAV using GPT-3.5 via the OpenAI API. Specifically, evaluating 1,000 examples took approximately 8 hours. While this might seem substantial, it is essential to highlight that PromptAV focuses not just on accuracy but also on providing detailed step-by-step stylometric explanations, which inherently requires intricate computations and, consequently, more time.
>
> We could also to batch-process texts. Nevertheless, OpenAI API has limits for request per minute and token per minute for each API key. To batch-process text, we can send multiple requests using multiple API keys. With adequate budget, we could scale-up the AV operations.
>
> Regarding the financial aspects, the expense associated with using PromptAV via the OpenAI API is indeed calculated at a rate of \\$0.004 per 1,000 tokens. For a dataset of 1,000 samples, the estimated cost came to around \\$10. This cost, we believe, is justifiable given the unique interpretability insights PromptAV provides, especially in critical applications like forensic analysis where the implications of the results are profound.
>
> We can also apply PromptAV on other open source LLMs such as LLaMA-2 and OPT in our future works. Applying PromptAV on these LLMs could significantly reduced the financial cost, and we could batch-process the text without any API limitations.
>
> ---
>
> ***W2/Q2**: The practicality of PromptAV is not fully considered, especially in experiment part. The paper uses an optimal threshold when reporting accuracy. Is this valid in real application? Can the authors provide more insights in the selection of the confidence score?*
>
> **[Response to W2/Q1]**
>
> Optimal Threshold Methodology: As you've rightly pointed out, the method of reporting using an optimal threshold was our initial approach to compare the results of different prompting methods. The intention was to establish a baseline comparison across varying methods in a controlled environment. In real-world applications, the threshold would likely require adjustment based on specific use-cases and available data.
>
> Practical Implications of Threshold: It is indeed a challenge to select an optimal threshold that would be universally valid. In our experiment applying PromptAV on the IMDb62 dataset, we found that the optimal threshold lies between 0.2 - 0.3. This range suggests that the model tends to generate a low confidence score, which provides significant insights into its functioning and decision-making. The low confidence score may be indicative of the model's cautious approach to attributing authorship, ensuring a reduced number of false positives in practical scenarios. This threshold may varied, depending on the complexity of the dataset, and the amount of writing sample observations the model made for each author.
>
> Selection of the Confidence Score: The confidence score selection was an empirical decision based on our preliminary experiments, intending to optimize the balance between false positives and false negatives. We have conducted an experiments to show the tradeoff on performance when selecting different threshold for the confident score. Note that experiments are conducted using PromptAV with zero-shot demonstration setting. We will include this experiment results in the appendix of our camera-ready version.
>
> | Confidence Score Threshold | Acc  |
> |-|--|
> | 0.1|0.506|
> | 0.2|0.587|
> | 0.3|0.579|
> | 0.4|0.578|
> | 0.5|0.569|
> | 0.6|0.55|
> | 0.7|0.53|
> | 0.8|0.519|
> | 0.9|0.5|
>
> Based on your feedback, we acknowledge that more rigorous experiments detailing the iterative process of threshold selection and its implications on real-world tasks will bolster the practicality of our approach. We will also add the above discussion into the updated version of our paper.
>
> ---
>
> ***W3/Q3**: The robustness of PromptAV is not analyzed, which is crucial for AV. Is PromptAV robust against a series of authorship obfuscation methods? If so, why?*
>
> **[Response to W3/Q3]**
>
> In our pursuit to understand PromptAV's potential robustness, we took a focused approach by testing it against an obfuscation dataset. We selected the suggested Mutant-X algorithm [1], a recognized method in the authorship obfuscation domain, and applied it to the IMDb62 dataset that was referenced in our paper. We apply PromptAV in zero-shot setting on the obfuscated dataset and benchmark against its performance on the original dataset. The experimental results are presented in the below table.
>
> | Model | Acc. |
> |-|-|
> |PromptAV| 0.587|
> |PromptAV_Obfuscated| 0.580|
> |PS+ Prompting| 0.536|
> |CoT| 0.524|
>
> The results reveal a negligible decline in performance when confronted with obfuscated text. Interestingly, the accuracy achieved, even with obfuscation, surpasses the results of zero-shot CoT and PS+ prompting on non-obfuscated text. This indicates a promising level of resilience in PromptAV against authorship obfuscation methods.
>
> While these preliminary results are encouraging, we acknowledge the need for more comprehensive testing against a broader range of obfuscation methods to assert PromptAV's robustness conclusively. We genuinely appreciate this insight and will work on expanding our experiments to present a more holistic understanding of PromptAV's capabilities in this area.
>
> ---
>
> ***Others**: Missing References*
>
> We will add the suggested reference and perform a more thorough literature review in the updated version of our paper.

---

### Meta-Review · Area_Chair_ctst · 2023-09-19

**Recommendation:** 3

**Metareview:**

The paper introduces PromptAV, a novel approach that uses Large Language Models (LLMs) for the task of authorship verification, incorporating chain-of-thought to prompt the verification process. Reviewers recognized the significance and interest in the paper's focus on authorship verification. While acknowledging the paper's contributions, reviewers also raised some valid concerns, which the authors addressed during the rebuttal stage and committed to resolving in the final version.

---

### Decision · Program_Chairs · 2023-10-07

**Decision:**

Accept-Findings

**Comment:**

The paper introduces PromptAV, a novel approach that uses Large Language Models (LLMs) for the task of authorship verification, incorporating chain-of-thought to prompt the verification process. Reviewers recognized the significance and interest in the paper's focus on authorship verification. While acknowledging the paper's contributions, reviewers also raised some valid concerns, which the authors addressed during the rebuttal stage and committed to resolving in the final version.